# POINCARÉ WASSERSTEIN AUTOENCODER

## ABSTRACT

This work presents the Poincaré Wasserstein Autoencoder, a reformulation of the recently proposed Wasserstein autoencoder framework on a non-Euclidean manifold, the Poincaré ball model of the hyperbolic space $\mathbb{H}^n$. By assuming the latent space to be hyperbolic, we can use its intrinsic hierarchy to impose structure on the learned latent space representations. We show that for datasets with latent hierarchies, we can recover the structure in a low-dimensional latent space. We also demonstrate the model in the visual domain to analyze some of its properties and show competitive results on a graph link prediction task.

## 1 INTRODUCTION

Variational Autoencoders (VAE) (17; 28) are an established class of unsupervised machine learning models, which make use of amortized approximate inference to parametrize the otherwise intractable posterior distribution. They provide an elegant, theoretically sound generative model used in various data domains.

Typically, the latent variables are assumed to follow a Gaussian standard prior, a formulation which allows for a closed form evidence lower bound formula and is easy to sample from. However, this constraint on the generative process can be limiting. Real world datasets often possess a notion of structure such as object hierarchies within images or implicit graphs. This notion is often reflected in the interdependence of latent generative factors or multimodality of the latent code distribution. The standard VAE posterior parametrizes a unimodal distribution which does not allow structural assumptions. Attempts at resolving this limitation have been made by either "upgrading" the posterior to be more expressive (27) or imposing structure by using various structured priors (34), (36). Furthermore, the explicit treatment of the latent space as a Riemannian manifold has been considered. For instance, the authors of (5) show that the standard VAE framework fails to model data with a latent spherical structure and propose to use a hyperspherical latent space to alleviate this problem. Similarly, we believe that for datasets with a latent tree-like structure, using a hyperbolic latent space, which imbues the latent codes with a notion of hierarchy, is beneficial.

There has recently been a number of works which explicitly make use of properties of non-Euclidean geometry in order to perform machine learning tasks. The use of hyperbolic spaces in particular has been shown to yield improved results on datasets which either present a hierarchical tree-like structure such as word ontologies (24) or feature some form of partial ordering (4). However, most of these approaches have solely considered deterministic hyperbolic embeddings.

In this work, we propose the Poincaré Wasserstein Autoencoder (PWA), a Wasserstein autoencoder (33) model which parametrizes a Gaussian distribution in the Poincaré ball model of the hyperbolic space $\mathbb{H}^n$. By treating the latent space as a Riemannian manifold with constant negative curvature, we can use the norm ranking property of hyperbolic spaces to impose a notion of hierarchy on the latent space representation, which is better suited for applications where the dataset is hypothesized to possess a latent hierarchy. We demonstrate this aspect on a synthetic dataset and evaluate it using a distortion measure for Euclidean and hyperbolic spaces. We derive a closed form definition of a Gaussian distribution in hyperbolic space $\mathbb{H}^n$ and sampling procedures for the prior and posterior distributions, which are matched using the Maximum Mean Discrepancy (MMD) objective. We also compare the PWA to the Euclidean VAE visually on an MNIST digit generation task as well quantitatively on a semi-supervised link prediction task.

The rest of this paper is structured as follows: we review related work in Section 2, give an overview of the mathematical tools required to work with Riemannian manifolds as well as define the notion

of probability distributions on Riemannian manifolds in Section 3. Section 4 describes the model architecture as well as the intuition behind the Wasserstein autoencoder approach. Furthermore, we derive a method to obtain samples from prior and posterior distributions in order to estimate the PWA objective. We present the performed experiments in and discuss the observed results in Section 5 and a summary of our results in Section 6.

## 2 RELATED WORK

**Amortized variational inference**  There has been a number of extensions to the original VAE framework (17). These extensions address various problematic aspects of the original model. The first type aims at improving the approximation of the posterior by selecting a richer family of distributions. Some prominent examples include the Normalizing Flow model (27) as well as its derivates (20), (16), (7). A second direction aims at imposing structure on the latent space by selecting structured priors such as the mixture prior (6), (34), learned autoregressive priors (36) or imposing informational constraints on the objective (12), (38). The use of discrete latent variables has been explored in a number of works (13) (36). The approach conceptually most similar to ours but with a hyperspherical latent space and a von-Mises variational distribution has been presented in (5).

**Hyperbolic geometry**  The idea of graph generation in hyperbolic space and analysis of complex network properties has been studied in (19). The authors of (24) have recently used both the Poincaré model and the Lorentz model (25) of the hyperbolic space to develop word ontology embeddings which carry hierarchical information encoded by the embedding norm. The general idea of treating the latent space as a Riemannian manifold has been explored in (2). A model for Bayesian inference for Riemannian manifolds relying on particle approximations has been proposed in (21). Finally the natural gradient method is a prime example for using the underlying information geometry imposed by the Fisher information metric to enhance learning performance (1).

Three concurrent works have explored an idea similar to ours. (22) propose to train a VAE with a hyperbolic latent space using the traditional evidence lower bound (ELBO) formulation. They approximate the ELBO using MCMC samples as opposed to our approach, which uses a Wasserstein formulation of the problem. (23) propose to use a *wrapped* Gaussian distribution to obtain samples on the Lorentz model of hyperbolic latent space. The samples are generated in Euclidean space using classical methods and then projected onto the manifold under a concatenation of a parallel transport and the exponential map at the mean. The authors of (10) also propose a similar approach but use an adversarial autoencoder model in their work instead.

## 3 HYPERBOLIC GEOMETRY

In this section, we briefly outline some of the concepts from differential geometry, which are necessary to formally define our model.

### 3.1 RIEMANNIAN GEOMETRY: A SHORT OVERVIEW

A *Riemannian manifold* is defined as a the tuple $(\mathcal{M}, g)$, where for every point $\mathbf{x}$ belonging to the manifold $\mathcal{M}$, a *tangent space* $\mathcal{T}_{\mathbf{x}}\mathcal{M}$ is defined, which corresponds to a first order local linear approximation of $\mathcal{M}$ at point $x$. The *Riemannian metric* $g$ is a collection of inner products $\langle \cdot | \cdot \rangle_{\mathbf{x}} : \mathcal{T}_{\mathbf{x}}\mathcal{M} \times \mathcal{T}_{\mathbf{x}}\mathcal{M} \to \mathbb{R}$ on the tangent spaces $\mathcal{T}_{\mathbf{x}}\mathcal{M}$. We denote by $\alpha(t) \in \mathcal{M}$ to be smooth curves on the manifold. By computing the speed vector $\dot{\alpha}(t)$ at every point of the curve, the Riemannian metric allows the computation of the curve length:

$$L_{(a,b)}(\alpha) = \int_a^b \sqrt{g_H(\dot{\alpha}(t), \dot{\alpha}(t))} dt$$

Given a smooth curve $\alpha(a, b) \to \mathcal{M}$, the distance is defined by the infimum over $\alpha(t)$: $d = \inf_{\alpha} L_{(a,b)}(\alpha)$. The smooth curves of shortest distance between two points on a manifold are called *geodesics*.

Given a point $\mathbf{x} \in \mathcal{M}$, the *exponential map* $\exp_{\mathbf{x}}(\mathbf{v}) : \mathcal{T}_{\mathbf{x}}\mathcal{M} \to \mathcal{M}$ gives a way map a vector $\mathbf{v}$ in the tangent space $\mathcal{T}_{\mathbf{x}}\mathcal{M}$ at point $\mathbf{x}$ to the corresponding point on the manifold $\mathcal{M}$. For the Poincaré ball model of the hyperbolic space, which is geodesically complete, this map is well defined on the whole tangent space $\mathcal{T}_{\mathbf{x}}\mathcal{M}$. The *logarithmic map* $\log_{\mathbf{x}}(\mathbf{v})$ is the inverse mapping from the manifold to the tangent space. The *parallel transport* $P_{\mathbf{x}_0 \to \mathbf{x}} : \mathcal{T}_{\mathbf{x}_0}\mathcal{M} \to \mathcal{T}_{\mathbf{x}}\mathcal{M}$ defines a linear isometry between two tangent spaces of the manifold and allows to move tangent vectors along geodesics.

## 3.2 POINCARÉ BALL

Hyperbolic spaces are one of three existing types of isotropic spaces: the Euclidean spaces with zero curvature, the spherical spaces with constant positive curvature and the hyperbolic spaces which feature constant negative curvature. The Poincaré ball is one of the five isometric models of the hyperbolic space. The model is defined by the tuple $(\mathcal{B}_n, g_H)$ where $\mathcal{B}_n$ is the open ball of radius 1, [1] $g_H$ is the hyperbolic metric and $g_E = I_n$ is the Riemannian metric on the flat Euclidean manifold.

$$\mathcal{B}_n = \{\mathbf{x} \in \mathbb{R}^n \mid ||\mathbf{x}|| < 1\} \quad g_H = \left(\frac{2}{1 - ||\mathbf{x}||^2}\right)^2 g_E = \lambda_{\mathbf{x}}^2 g_E$$

The geodesic distance on the Poincaré ball is given by

$$d(\mathbf{x}, \boldsymbol{\mu}) = \operatorname{arccosh}\left(1 + 2\frac{||\mathbf{x} - \boldsymbol{\mu}||^2}{(1 - ||\mathbf{x}||^2)(1 - ||\boldsymbol{\mu}||^2)}\right) \tag{1}$$

## 3.3 GYROVECTOR SPACES FRAMEWORK AND OPERATORS

In order to perform arithmetic operations on the Poincaré ball model, we rely on the concept of gyrovector spaces, which is a generalization of Euclidean vector spaces to models of hyperbolic space based on Möbius transformations. First proposed by (35), they have been recently used to describe typical neural network operations in the Poincaré ball model of hyperbolic space (9). In order to perform the reparametrization in hyperbolic space, we use the gyrovector addition and Hadamard product defined as a diagonal matrix-gyrovector multiplication. Furthermore, we make use of the exponential $\exp_{\boldsymbol{\mu}}$ and logarithm $\log_{\boldsymbol{\mu}}$ map operators in order to map points onto the manifold and perform the inverse mapping back to the tangent space. The Gaussian decoder network is symmetric to the encoder network.

# 4 MODEL

## 4.1 GAUSSIAN DISTRIBUTION IN $\mathbb{H}^n$

The Gaussian distribution is a common choice of prior for VAE style models. Similarly to the VAE, we can select a generalization of the Gaussian distribution in the hyperbolic space as prior for our model. In particular, we choose the maximum entropy generalization of the Gaussian distribution (26) on the Poincaré ball model. The Gaussian probability density function in hyperbolic space is defined via the Fréchet mean $\boldsymbol{\mu}$ and dispersion parameter $\sigma > 0$, analogously to the density in the Euclidean space.

$$\mathcal{N}_H(\mathbf{x}|\boldsymbol{\mu}, \sigma) = \frac{1}{Z(\sigma)} e^{-\frac{d^2(\mathbf{x}, \boldsymbol{\mu})}{2\sigma^2}} \tag{2}$$

The main difference compared to Euclidean space is the use of the geodesic distance $d(\mathbf{x}, \boldsymbol{\mu})$ in the exponent and a different dispersion dependent normalization constant $Z(\sigma)$ which accounts for the underlying geometry. In order to compute the normalization constant, we use hyperbolic

---

[1]this can be generalized to radius $\frac{1}{\sqrt{c}}$ for curvature $c$. Throughout this paper, we assume the Poincaré ball radius to be $c = 1$ and omit it from the notation.

polar coordinates where $r = d(\mathbf{x}, \boldsymbol{\mu})$ is the geodesic distance between the $\mathbf{x}$ and $\boldsymbol{\mu}$. This allows the decomposition of $Z(\sigma)$ into radial and angular components.

$$Z(\sigma) = Z_\alpha(\sigma)Z_r(\sigma) = \text{Vol}(\mathbb{S}^{n-1}) \times \int_0^\infty e^{-\frac{r^2}{2\sigma^2}} \sinh^{n-1}(r)dr$$

We derive the closed form of the normalization constant in appendix A. For a two-dimensional space, the normalization constant is given by (30):

$$Z(\boldsymbol{\sigma}) = 2\pi \sqrt{\frac{\pi}{2}} \sigma e^{\frac{\sigma^2}{2}} \text{erf}\left(\frac{\boldsymbol{\sigma}}{\sqrt{2}}\right)$$

**Dispersion representation**    The closed form of the hyperbolic Gaussian distribution (2) is only defined for a scalar dispersion value. This can be a limitation on the expressivity of the learned representations. However, the variational family which is implicitly given by the hyperbolic reparametrization allows for vectorial or even full covariance matrix representations, which can be more expressive. Since the maximum mean discrepancy can be estimated via samples, we do not require a closed form definition of the posterior density as is the case with training using the evidence lower bound. This allows the model to learn richer latent space representations.

## 4.2    Model architecture

Our model mimics the general architecture of a variational autoencoder. The encoder parametrizes the posterior variational distribution $q_\phi(\mathbf{z}|\mathbf{x})$ and the decoder parametrizes the unit variance Gaussian likelihood $p_\theta(\mathbf{x}|\mathbf{z})$. In order to accomodate the change in the underlying geometry of the latent space, we introduce the maps into hyperbolic space and back to the tangent space. Both the encoder and decoder network consist of three fully-connected layers with ReLU activations. We use the recently proposed hyperbolic feedforward layer (9) for the encoding of the variational family parameters $(\boldsymbol{\mu}_H, \boldsymbol{\sigma})$. For the decoder $f_\theta(\mathbf{x}|\mathbf{z})$, we use the logarithm map at the origin $\log_{\mathbf{0}}(\mathbf{z})$ to map the posterior sample $\mathbf{z}$ back into the tangent space.

**Mean and variance parametrization**    In order to obtain posterior samples in hyperbolic space, the parametrization of the mean uses a hyperbolic feedforward layer $(W, \mathbf{b}_H)$ as the last layer of the encoder network (proposed in (9)). The weight matrix parameters are Euclidean and are subject to standard Euclidean optimization procedures (we use Adam (15)) while the bias parameters are hyperbolic, requiring the use of Riemannian stochastic gradient descent (RSGD) (3). The outputs of the underlying Euclidean network $\mathbf{h}$ are projected using the exponential map at the origin and transformed using the hyperbolic feedforward layer map where $\varphi_h$ is the hyperbolic nonlinearity [2]:

$$f_h(\mathbf{h}) = \varphi_h(W^\otimes \exp_{\mathbf{0}}(\mathbf{h}) \oplus \mathbf{b_h}) \quad \varphi_h(\mathbf{x}) = \exp_{\mathbf{0}}(\varphi(\log_{\mathbf{0}} \mathbf{x}))$$

## 4.3    Hyperbolic reparametrization trick

The reparametrization trick is a common method to make the sampling operation differentiable by using a differentiable function $g(\epsilon, \theta)$ to obtain a reparametrization gradient for backpropagation through the stochastic layer of the network. For the location-scale family of distributions, the reparametrization function $g(\epsilon, \theta)$ can be written as $\mathbf{z} = \boldsymbol{\mu} + \boldsymbol{\sigma} \odot \epsilon$ in the Euclidean space where $\epsilon \sim \mathcal{N}(\mathbf{0}, \mathbf{I})$. We adapt the reparametrization trick for the Gaussian distribution in the hyperbolic space by using the framework of gyrovector operators. We obtain the posterior samples for the parametrized mean $\boldsymbol{\mu}_H(\mathbf{x})$ and dispersion $\boldsymbol{\sigma}(\mathbf{x})$ using the following relation:

$$\mathbf{z} = \mu_H(\mathbf{x}) \oplus \text{diag}(\sigma(\mathbf{x}))^\otimes \epsilon \tag{3}$$

We can motivate the reparametrization (3) with the help of Fig. 1, which depicts the reparametrization in a graphical fashion. In a first step, we sample $\epsilon$ from the hyperbolic standard prior $\epsilon \sim \mathcal{N}_H(0, 1)$ using a rejection sampling procedure we describe in Algorithm 1. The samples are projected to the

---

[2]see Appendix C for the operator definitions

tangent space using the logarithm map $\log_{\mathbf{0}}$ at the origin, where they are scaled using the dispersion parameter. The scaled samples are then projected back to the manifold using the exponential map $\exp_{\mathbf{0}}$ and translated using $\boldsymbol{\mu}$.

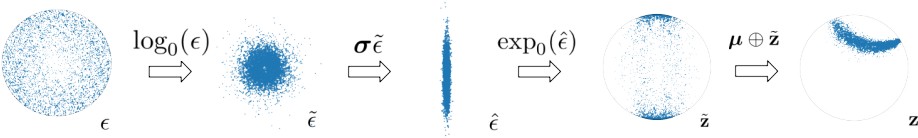

Figure 1: Hyperbolic Gaussian reparametrization

### 4.3.1 Sampling from prior in hyperbolic space

We choose the hyperbolic standard prior $\mathcal{N}_H(0, I)$ as prior $p(\mathbf{z})$. In order to generate samples from the standard prior, we use an approach based on the volume ratio of spheres in $\mathbb{H}^d$ to obtain the quasi-uniform samples on the Poincaré disk (19) and subsequently use a rejection sampling procedure to obtain radius samples. We use the quasi-uniform distribution

$$g(r) = \alpha e^{\alpha(r-R)}$$

as a proposal distribution for the radius. Using the decomposition into radial and angular components, we can sample a direction from the unit sphere uniformly and simply scale using the sampled radius to obtain the samples from the prior. An alternative choice of prior is the *wrapped Gaussian* distribution. The samples are obtained by sampling $\tilde{\mathbf{z}} \sim \mathcal{N}(0, 1)$ in the tangent space and projecting them onto the latent space manifold. Empirically, we have found the prior obtained via the rejection sampling procedure and the exponential map prior to perform similarly in the context of the PWA objective. A comparison of samples from both distributions is presented in Appendix D.

### 4.4 Optimization

**Evidence Lower Bound**   The variational autoencoder relies on the evidence lower bound (ELBO) reformulation in order to perform tractable optimization of the Kullback-Leibler divergence (KLD) between the true and approximate posteriors. In the Euclidean VAE formulation, the KLD integral has a closed-form expression, which simplifies the optimization procedure considerably.

The definition of the evidence lower bound can be extended to non-Euclidean spaces by using the following formulation with the volume element of the manifold $d\mathrm{vol}_{g_H}$ induced by the Riemannian metric $g_H$.

$$\log p(x) = \log \int_{\mathcal{M}} p(x|z)p(z) d\mathrm{vol}_{g_H} = \log \int_{\mathcal{M}} \frac{p(x|z)p(z)}{q_\phi(z|x)} q_\phi(z|x)p(z) d\mathrm{vol}_{g_H}$$
$$\geq \int_{\mathcal{M}} \log \frac{p(x|z)p(z)}{q_\phi(z|x)} q_\phi(z|x) d\mathrm{vol}_{g_H} = \mathbb{E}_{z \sim q_\phi}[\log p(x|z) + \log p(z) - \log q_\phi(z|x)]$$

By substituting the hyperbolic Gaussian (2) into (4) we obtain the following expressions for $\mathbb{E}_{q_\phi(\mathbf{z}')} \log q_\phi(z|x)$:

$$\mathbb{E}_{q_\phi(\mathbf{z}')}[\log q_\phi(z|x)] = \mathbb{E}_{q_\phi(\mathbf{z}')}[\log \frac{1}{Z(\sigma)} - \frac{d^2(\mathbf{x}, \mu)}{\sigma^2}] = \mathrm{const} - \mathbb{E}_{q_\phi(\mathbf{z}')}[\log^2(\mathbf{x} + \sqrt{\mathbf{x}^2 - 1})]$$

Due to the nonlinearity of the geodesic distance in the exponent, we cannot derive a closed form solution of the expectation expression $\mathbb{E}_{q_\phi(\mathbf{z})}[\log q_\phi(\mathbf{z})]$. One possibility is to use a Taylor expansion of the first two moments of the expectation of the squared logarithm. This is however problematic from a numerical standpoint due to the small convergence radius of the Taylor expansion. The

ELBO can be approximated using Monte-Carlo samples, as is done in (22). We have considered this approach to be suboptimal due to large variance associated with one-sample MC approximations of the integral.

**Wasserstein metric**   In order to circumvent the high variance associated with the MC approximation we propose to use a Wasserstein Autoencoder (WAE) formulation of the variational inference problem. The authors of the WAE framework propose to solve the *optimal transport* problem for matching distributions in the latent space instead of the more difficult problem of matching the data distribution $p(\mathbf{x})$ to the distribution generated by the model $p_{\mathbf{y}}(\mathbf{z})$ as is done in the generative adversarial network (GAN) literature. Kantorovich's formulation of the optimal transport problem is given by:

$$W_c(p_x, p_g) = \inf_{\Gamma \in p(\mathbf{x} \sim p_{\mathbf{x}}, \mathbf{y} \sim p_{\mathbf{y}})} \mathbb{E}_{(\mathbf{x}, \mathbf{y}) \sim \Gamma}[c(\mathbf{x}, \mathbf{y})] \tag{4}$$

where $c(\mathbf{x}, \mathbf{y})$ is the cost function, $p(\mathbf{x} \sim p_{\mathbf{x}}, \mathbf{y} \sim p_{\mathbf{y}})$ is the set of joint distributions of the variables $\mathbf{x} \sim p_{\mathbf{x}}$ and $y \sim p_{\mathbf{y}}$. Solving this problem requires a search over all possible couplings $\Gamma$ of the two distributions which is very difficult from an optimization perspective. The issue is circumvented in a WAE model as follows. The generative model of a variational autoencoder is defined by two steps. First we sample a latent variable $\mathbf{z}$ from the latent space distribution $p(\mathbf{z})$. In a second step, we map it to the output space using a deterministic parametric decoder $f_\theta(\mathbf{x}|\mathbf{z})$. The resulting density is given by:

$$p(\mathbf{x}) = \int_{\mathcal{Z}} f_\theta(\mathbf{x}|\mathbf{z}) p(\mathbf{z})$$

Under this model, the optimal transport cost (5) takes the following simpler form due to the fact that the transportation plan factors through the map $f_\theta$.

$$W_c(p_{\mathbf{x}}, p_{\mathbf{y}}) = \inf_{\Gamma \in p(\mathbf{x} \sim p_{\mathbf{x}}, \mathbf{y} \sim p_{\mathbf{y}})} \mathbb{E}_{(\mathbf{x}, \mathbf{y}) \sim \Gamma}[c(\mathbf{x}, \mathbf{y})] = \inf_{q: q_\phi(\mathbf{z}) = p(\mathbf{z})} \mathbb{E}_{p_{\mathbf{x}}} \mathbb{E}_{q_\phi}[c(\mathbf{x}, f_\theta(\hat{\mathbf{x}}|\mathbf{z}))]$$

The optimization procedure is over the encoders $q_\phi(\mathbf{x})$ instead of the couplings between $p_{\mathbf{x}}$ and $p_{\mathbf{y}}$. The WAE objective is derived from the optimal transport cost (5) by relaxing the constraint on the posterior $q$. The constraint is relaxed by using a Lagrangian multiplier and an appropriate divergence measure.

$$\mathcal{L}_{\text{WAE}} = \inf_{q_\phi(\mathbf{z}|\mathbf{x}) \in \mathcal{Q}} \mathbb{E}_{p(\mathbf{x})} \mathbb{E}_{q(\mathbf{z}|\mathbf{x})} (\log p(\mathbf{x}|\mathbf{z})) + \beta D_{\text{MMD}} \tag{5}$$

The Maximum Mean Discrepancy (MMD) metric with an appropriate positive definite RKHS [3] kernel is an example of such a divergence measure. MMD is known to perform well when matching high-dimensional standard normal distributions (11). MMD is a metric on the space of probability distributions under the condition that the selected RKHS kernel is *characteristic*. Geodesic kernels are generally not positive definite, however it has been shown that the Laplacian kernel $k(\mathbf{x}, \mathbf{y}) = \exp(-\lambda(d_H(\mathbf{x}, \mathbf{y})))$ is positive definite if the metric of the underlying space is conditionally negative definite (8). In particular, this holds for hyperbolic spaces (14). In practice, there is a high probability that a geodesic RBF kernel is also positive definite depending on the dataset topology (8). We choose the Laplacian kernel as it also features heavier tails than the Gaussian RBF kernel, which has a positive effect on outlier gradients (33). The MMD loss function is defined over two probability measures $p$ and $q$ in an RKHS unit ball $\mathcal{F}$ as follows:

$$D_{\text{MMD}}(p, q_\phi) = || \int_{\mathcal{Z}} k(\mathbf{z}, \cdot) dp(\mathbf{z}) - \int_{\mathcal{Z}} k(\mathbf{z}, \cdot) dq_\phi(\mathbf{z}) ||_{\mathcal{F}} \tag{6}$$

There exists an unbiased estimator for $D_{\text{MMD}}(p, q_\phi)$. A finite sample estimate can be computed based on minibatch samples from the prior $\mathbf{z} \sim p(\mathbf{z})$ via the rejection sampling procedure described

---

[3]RKHS = Reproducing Kernel Hilbert Space

Table 1: Average distortion measure

| Dataset | Metric | T-SNE | $\mathcal{N}$-VAE | PWA |
|---------|--------|-------|-------------------|-----|
| Synthetic trees | Avg | 0.73 | 0.82 | **0.49** |

in Appendix A and the approximate posterior samples $\bar{\mathbf{z}} \sim q_\phi(\mathbf{z})$ obtained via the hyperbolic reparametrization:

$$D_{\text{MMD}}^{(B)}(p(\mathbf{z}), q_\phi(\mathbf{z})) = \frac{\lambda}{n(n-1)} \sum_{i \neq j} k(\mathbf{z}_i, \mathbf{z}_j) + \frac{\lambda}{n(n-1)} \sum_{i \neq j} k(\bar{\mathbf{z}}_i, \bar{\mathbf{z}}_j) - \frac{2\lambda}{n^2} \sum_{i,j} k(\mathbf{z}_i, \bar{\mathbf{z}}_j) \quad (7)$$

**Parameter updates** The hyperbolic geometry of the latent space requires us to perform Riemannian stochastic gradient descent (RSGD) updates for a subset of the model parameters, specifically the bias parameters of $\boldsymbol{\mu}$. We perform full exponential map updates using gyrovector arithmetic for the gradients with respect to the hyperbolic parameters similar to (9) instead of using a retraction approximation as in (24). In order to avoid numerical problems at the origin and far away from the origin of the Poincaré ball, we perturb the operands if the norm is close to 0 or 1 respectively. The Euclidean parameters are updated in parallel using the Adam optimization procedure (15).

## 5 EXPERIMENTS

### 5.1 DISTORTION OF TREE-STRUCTURED DATA

To determine the capability of the model to retrieve an underlying hierarchy, we have setup two experiments in which we measure the average distortion of the respective latent space embeddings. We measure the distortion between the input and latent spaces using the following distortion metric, where subscript U denotes the distances in the input space and V the distances in the latent space.

$$d_{avg} = \frac{|d_V(f(\mathbf{a}), f(\mathbf{b})) - d_U(\mathbf{a}, \mathbf{b})|}{d_U(\mathbf{a}, \mathbf{b})}$$

**Noisy trees** The first dataset is a set of synthetically generated noisy binary trees. The vertices of the main tree are generated from a normal distribution where the mean of the child nodes corresponds to the parent sample $\mathbf{x}_i = \mathcal{N}(\mathbf{x}_{p(i)}, \sigma_i)$ and $p(i)$ denotes the index of the parent node. In addition to the main tree, we add $K$ noise samples $\tilde{\mathbf{x}}_j = \mathcal{N}(\mathbf{x}_i, \sigma_j)$ for every vertex. The dataset $\mathcal{D} = \{[\mathbf{x}_i, \tilde{\mathbf{x}}_j]\}_{i,j}$ is a concatenation of $\mathbf{x}_i$ and $\tilde{\mathbf{x}}_j$. To encourage a good embedding in a hyperbolic space, we enforce the norms of the tree vertices to grow monotonously with the depth of the tree by rejecting samples whose norms are smaller than the norm of the parent vertices. We have trained our model on 100 generated trees for 100 epochs. The tree vertex variance was set to $\sigma_i = 1$ and the noise variance to $\sigma_j = 0.1$. We have also normalized the generated vertices to zero mean and unit variance. Table 1 compares the distortion values of the test set latent space embeddings obtained by using the Euclidean VAE model compared to the PWA model. We can see that the PWA model shows less distortion when embedding trees into the latent space of dimension $d = 2$, which confirms our hypothesis that a hyperbolic latent space is better suited to data with latent hierarchies. As a reference, we provide the distortion scores obtained by the classical T-SNE (37) dimensionality reduction technique.

### 5.2 MNIST

In this experiment, we apply our model to the task of generating MNIST digits in order to get an intuition for the properties of the latent hyperbolic geometry. In particular, we are interested in the visual distibution of the latent codes in the Poincaré disk latent space. While the MNIST latent space is not inherently hierarchically structured - there is no obvious norm ranking that can be imposed - we can use it to compare our model to the Euclidean VAE approach. We train the models on dynamically

binarized MNIST digits and evaluate the generated samples qualitatively as well as quantitatively via the reconstruction error scores. We can observe in Appendix B that the samples present a deteriorating quality as the dimensionality increases despite the lower reconstruction error. This can be explained by the issue of dimension mismatch between the selected latent space dimensionality $d_z$ and the intrinsic latent space dimensionality $d_I$ documented in (29) and can be alleviated by an additional $p$-norm penalty on the variance. We have not observed a significant improvement by applying the L2-penalty for higher dimensions. We have also performed an experiment using a two-dimensional latent space. We can observe that the structure imposed by the Poincaré disk pushes the samples towards the outside of the disk. This observation can be explained by the fact that hyperbolic spaces grow exponentially. In order to generate quality samples using the prior, some overlap is required with the approximate posterior in the latent space. The issue is somewhat alleviated in higher dimensions as the distribution shifts towards the ball surface.

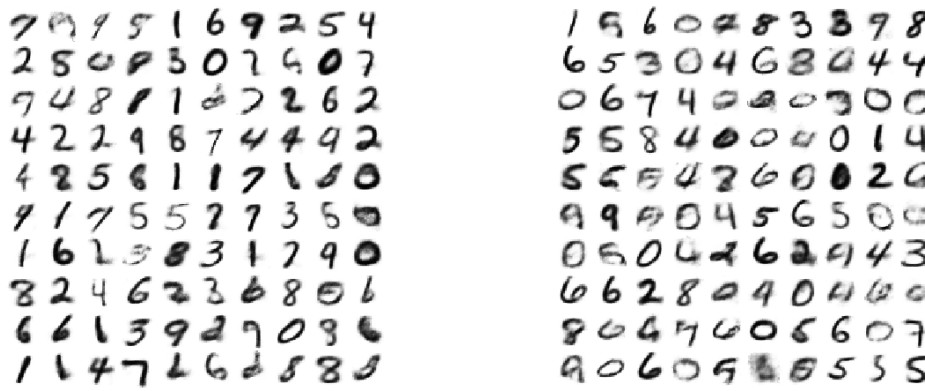

Figure 2: A comparison of the Euclidean VAE (left) and PWA samples (right), $|\mathbf{z}| = 5$

## 5.3 LINK PREDICTION ON CITATION NETWORKS

In this experiment, we aim at exploring the advantages of using a hyperbolic latent space on the task of predicting links in a graph. We train our model on three different citation network datasets: Cora, Citeseer and Pubmed (32). We use the Variational Graph Auto-Encoder (VGAE) framework (18) and train the model in an unsupervised fashion using a subset of the links. The performance is measured in terms of average precision (AP) and area under curve (AUC) on a test set of links that were masked during training. Table 1 shows a comparison to the baseline with a Euclidean latent space ($\mathcal{N}$-VGAE), showing improvements on the Cora and Citeseer datasets. We also compare our results to the results obtained using a hyperspherical autoencoder ($\mathcal{S}$-VGAE) (5). It should be noted that we have used a smaller dimensionality for the hyperbolic latent space (16 vs 64 and 32 for the Euclidean and hyperspherical cases respectively), which could be attributed to the fact that a dataset with a hierarchical latent manifold requires latent space embeddings of smaller dimensionality to efficiently encode the information (analogously to the results of (24)). We can observe that the PWA outperforms the Euclidean VAE on two of the three datasets. The hyperspherical graph autoencoder ($\mathcal{S}$-VGAE) outperforms our model. One hypothesis which explains this is the fact that the structure of the citation networks has a tendency towards a positive curvature rather than a negative one. It is worth noting that it is not entirely transparent whether the use of Graph Convolutional Networks (18), which present a very simple local approximation of the convolution operator on graphs, allows to preserve the curvature of the input data.

Table 2: Performance on link prediction datasets

| Dataset | Metric | $\mathcal{N}$-VGAE | $\mathcal{S}$-VGAE | PWA |
|---------|--------|-----------|-----------|-----|
| Cora | AUC | $92.7_{\pm.2}$ | $94.1_{\pm.1}$ | $93.9_{\pm.2}$ |
|      | AP  | $93.2_{\pm.4}$ | $94.1_{\pm.3}$ | $93.2_{\pm.2}$ |
| Citeseer | AUC | $90.3_{\pm.5}$ | $94.7_{\pm.2}$ | $92.2_{\pm.2}$ |
|          | AP  | $91.5_{\pm.5}$ | $95.2_{\pm.2}$ | $91.8_{\pm.2}$ |
| Pubmed | AUC | $97.1_{\pm.0}$ | $96.0_{\pm.1}$ | $95.9_{\pm.2}$ |
|        | AP  | $97.1_{\pm.0}$ | $96.0_{\pm.2}$ | $96.3_{\pm.2}$ |

The "Model" header spans the $\mathcal{N}$-VGAE, $\mathcal{S}$-VGAE, and PWA columns.

## 6 CONCLUSION

We have presented an algorithm to perform amortized variational inference on the Poincaré ball model of the hyperbolic space. The underlying geometry of the hyperbolic space allows for an improved performance on tasks which exhibit a partially hierarchical structure. We have discovered certain issues related to the use of the MMD metric in hyperbolic space. Future work will aim to circumvent these issues as well as extend the current results. In particular, we hope to demonstrate the capabilities of our model on more tasks hypothesized to have a latent hyperbolic manifold and explore this technique for mixed curvature settings.

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

## A    PRIOR REJECTION SAMPLING

---
**Algorithm 1**: prior rejection sampling
---

**Input:** maximum support radius $r_{max}$, dimensionality $d$, quasi-uniform $\alpha$ parameter, hyperbolic prior likelihood $\mathcal{N}_{\mathcal{H}}^{(r)}(r|0,1)$
**Result:** $n$ samples from prior $p(\mathbf{z})$
**while** $i < n$ **do**
    sample $\tilde{\boldsymbol{\varphi}} \sim \mathcal{N}(\mathbf{0}, \mathbf{I}_d)$;
    compute direction on the unit sphere $\tilde{\boldsymbol{\varphi}} = \frac{\tilde{\boldsymbol{\varphi}}}{||\tilde{\boldsymbol{\varphi}}||}$;
    sample $u \sim \mathcal{U}(0,1)$;
    get uniform radius samples $r_i \in [0, r_{max}]$ via ratio of hyperspheres;
    $r_i = (u * r_{max}^d)^{\frac{1}{d}}$;
    evaluate $p(\mathbf{x}_i) = f_r(r_i)$;
    $M = \max(p_i)$;
    $g(r) = \alpha e^{\alpha(r - r_{max})}$;
    sample $u \sim \mathcal{U}(0,1)$;
    **if** $u < \frac{p_i}{M * g(r_i)}$ **then**
        | accept sample $\mathbf{x}_i$;
    **else**
        | reject sample;
    **end**
**end**
**Output:**  prior samples $z = r\varphi$

## B    NORMALIZATION CONSTANT DERIVATION

We can derive the normalization constant $Z(\sigma)$ for the Gaussian distribution in $\mathbb{H}^n$ for curvature $c = -1$ by using the hyperbolic polar coordinates. The integral form is given by (31):

$$Z(\sigma) = Z_r(\sigma) Z_\alpha(\sigma) = \text{Vol}(\mathbb{S}^{n-1}) \times \int_0^\infty e^{-\frac{r^2}{2\sigma^2}} \sinh^{n-1}(r) dr$$

The normalization constant can be factorized into the radius and the angle part. The volume of a unit hypersphere is given by:

$$Z_\alpha(\sigma) = \text{Vol}(\mathbb{S}^{n-1}) = \frac{\pi^{\frac{n-1}{2}}}{\Gamma\left(\frac{n-1}{2} + 1\right)}$$

We can derive a closed form for the normalization constant as follows:

$$Z_r(\sigma) = \int_0^\infty e^{-\frac{r^2}{2\sigma^2}} \sinh^{n-1}(r) dr$$

$$= \int_0^\infty e^{-\frac{r^2}{2\sigma^2}} \left(\frac{e^r - e^{-r}}{2^{n-1}}\right)^{n-1} dr$$

$$= \int_0^\infty e^{-\frac{r^2}{2\sigma^2}} \sum_{k=0}^{n-1} \binom{n-1}{k}(-1)^k \left(\frac{e^{(n-1-k)r}e^{(-rk)}}{2^{n-1}}\right) dr$$

$$= \frac{1}{2^{n-1}} \sum_{k=0}^{n-1} \binom{n-1}{k}(-1)^k \int_0^\infty e^{-\frac{r^2}{2\sigma^2}} e^{(n-1-k)r}e^{(-rk)} dr$$

$$= \frac{1}{2^{n-1}} \sum_{k=0}^{n-1} \binom{n-1}{k}(-1)^k \int_0^\infty e^{-\left(\frac{1}{2\sigma^2}r^2 - (n-1-2k)r\right)} dr$$

We use the following identity for the solution of the definite integral:

$$\int_0^\infty e^{-(ax^2+bx)} dx = \frac{1}{2}\sqrt{\frac{\pi}{a}} e^{\left(\frac{b^2-4ac}{4a}\right)} \mathrm{erfc}\left(\frac{b}{2\sqrt{a}}\right)$$

Setting $a = \frac{1}{2\sigma^2}$, $b = 2k+1-n$, $c = 0$, we obtain

$$Z_r(\sigma) = \frac{1}{2^{n-1}} \sum_{k=0}^{n-1} \binom{n-1}{k}(-1)^k \sqrt{\frac{\pi}{2}}\sigma e^{\frac{(2k+1-n)^2\sigma^2}{2}} \mathrm{erfc}\left(\frac{(2k+1-n)\sigma}{\sqrt{2}}\right)$$

where **erfc** is the complementary error function.

## C  LIST OF GYROVECTOR OPERATIONS

In this list of gyrovector operations and throughout this paper, we assume the Poincaré ball radius to be $c = 1$ and omit it from the notation.

Gyrovector addition:

$$\mathbf{x} \oplus \mathbf{y} = \frac{(1 + 2\langle \mathbf{x}, \mathbf{y}\rangle + ||\mathbf{y}||^2)\mathbf{x} + (1 - ||\mathbf{x}||^2)\mathbf{y}}{1 + 2\langle \mathbf{x}, \mathbf{y}\rangle + ||\mathbf{x}||^2||\mathbf{y}||^2}$$

Matrix-gyrovector product:

$$M^\otimes \mathbf{x} = \tanh\left(\frac{||M\mathbf{x}||}{||x||}\mathrm{arctanh}(||\mathbf{x}||)\right) \frac{M\mathbf{x}}{||M\mathbf{x}||}$$

Exponential map:

$$\exp_{\mathbf{x}}(\mathbf{v}) = \mathbf{x} \oplus \left(\tanh\left(\frac{\lambda_{\mathbf{x}}||\mathbf{v}||}{2}\right) \frac{\mathbf{v}}{||\mathbf{v}||}\right)$$

Logarithm map:

$$\log_{\mathbf{x}}(\mathbf{v}) = \frac{2}{\lambda_{\mathbf{x}}}\mathrm{arctanh}(|| - \mathbf{x} \oplus \mathbf{v}||)\frac{-\mathbf{x} \oplus \mathbf{v}}{|| - \mathbf{x} \oplus \mathbf{v}||}$$

Parallel transport:

$$P_{\mathbf{x}_0 \to \mathbf{x}}(\mathbf{v}) = \log_{\mathbf{x}}(\mathbf{v} \oplus \exp_{\mathbf{x}_0}(\mathbf{v})) = \frac{\lambda_{\mathbf{x}_0}}{\lambda_{\mathbf{x}}}\mathbf{v}$$

# D  HYPERBOLIC GAUSSIAN SAMPLES

This section presents a comparison of samples obtained from the hyperbolic Gaussian and the wrapped Gaussian distributions.

The means and variances are given as follows. $(\boldsymbol{\mu}_1, \boldsymbol{\sigma}_1) = ((0.0, 0.0), (1.0, 1.0))$, $(\boldsymbol{\mu}_2, \boldsymbol{\sigma}_2) = ((0.6, 0.0), (1.0, 1.0))$, $(\boldsymbol{\mu}_3, \boldsymbol{\sigma}_3) = ((0.6, 0.4), (1.0, 1.0))$, $(\boldsymbol{\mu}_4, \boldsymbol{\sigma}_4) = ((0.6, 0.4), (0.7, 0.3))$, $(\boldsymbol{\mu}_5, \boldsymbol{\sigma}_5) = ((0.6, 0.4), (0.1, 0.4))$

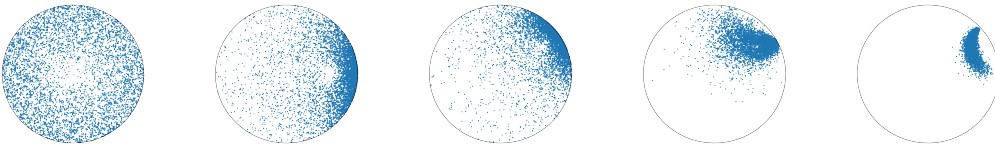

Figure 3: Hyperbolic Gaussian samples

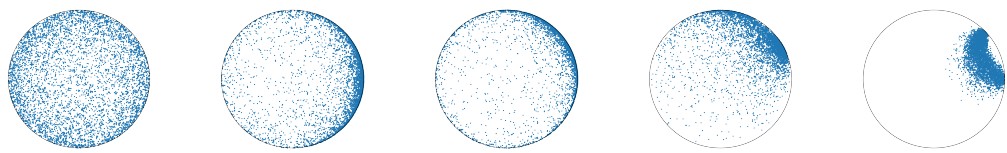

Figure 4: Wrapped Gaussian samples

# E    MNIST VISUAL SAMPLES

Figure 5: Euclidean VAE samples $d \in \{5, 10, 20\}$, training reconstruction error $L \in \{109.01, 94.58, 93.36\}$

Figure 6: Poincaré WAE samples $d \in \{5, 10, 20\}$, training reconstruction error $L \in \{95.01, 69.70, 58.58\}$

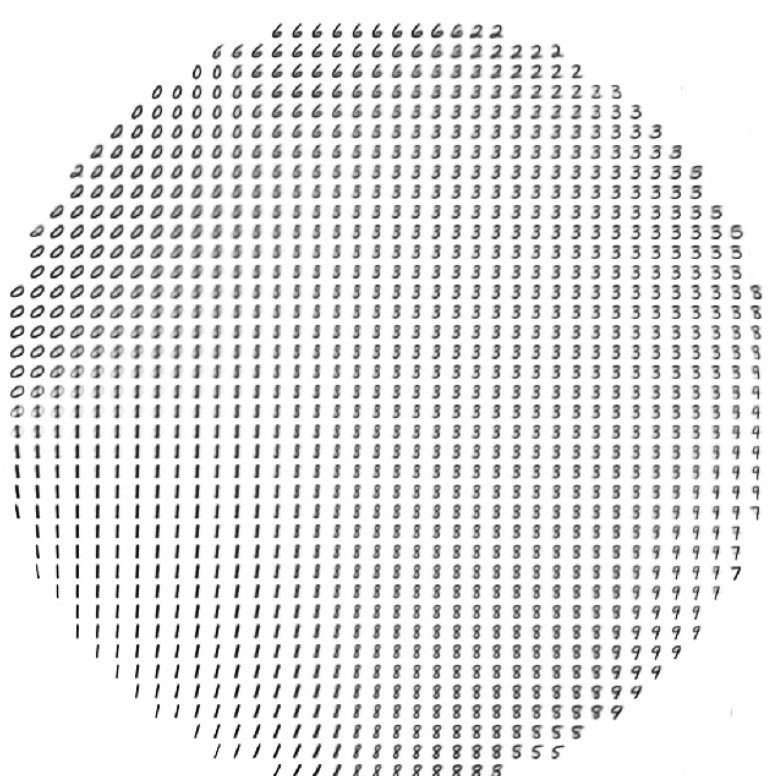

Figure 7: MNIST samples from two-dimensional hyperbolic latent space

