# OpenReview forum: "Poincaré Wasserstein Autoencoder"
_ICLR.cc/2020/Conference — Reject_

### Official Review · AnonReviewer2 · 2019-10-21
**Official Blind Review #2**

**Rating:** 3

**Review:**

*Paper summary*

The authors endow a variational auto-encoder with a hyperbolic space for the latent space. This enables them to model hierarchically ranked relationships in the data in a natural way. They use a Wasserstein formulation instead of the standard ELBO, which they claim reduces the variance of the gradients during training.

*Paper decision*

Thanks for an interesting paper, I enjoyed reading it. While there are parts of this paper I like very much and I think it is technically sound, I feel the experiments section is lacking somewhat (see comments below). I therefore am recommending a weak reject; although, upon a favourable rebuttal would happily upgrade this position.

*Supporting arguments*

- In the introduction is well written and the motivation for the introduction of a hyperbolic later space is well laid out.
- The section on Hyperbolic geometry is very well written indeed and presents this non-trivial material in a very simple to understand manner. Maybe what it lacks a bit is motivation for a machine learning audience, for why these mathematical tools from differential geometry would be useful or needed by the community to tackle presenting machine learning problems.
- The structure of the method is well laid out and every hyperbolic version of an auto-encoder operation is explained. I would prefer to see methods from cited works, which are deployed in this paper, written in the paper instead of just referred to, so that I don’t have to search through the cited works to find out how a key operation works.
- I think the method itself is technically sound.
- I am not sure about the novelty of the method, seeing that there is a very similar paper from earlier this year (Mathieu et al. 2019).
- The experiments are lacking a little. I do not feel that they show a clear reason why a hyperbolic latent space should be used and compared to standard methods, they do not perform any. better. It would have been nice to have a more structured discussion on the merits of the method.

*Questions/notes for the authors*

- Section 3.2, please explain the significance of \lambda_x^c
- Please explain in the related work and/or introduce in more detail than you already have the differences and similarities between this work and your nearest neighbour “Hierarchical representations with poincaré variational auto-encoders” by Mathieu et al. (2019). I would like to know the advantages/disadvantages of using a Wasserstein formulation instead of a sampled ELBO.
- In section 3.3, please point readers forward to the Appendix for a list of gyrovector operations.
- Please use equation numbers on all equations.
- In your section on “Dispersion representation” you state: “Since the maximum mean discrepancy can be estimated via samples, we do not require a closed form definition of the posterior density as is the case with training using the evidence lower bound. This allows the model to learn richer latent space representations.” Why exactly does not using a closed form posterior density lead to a richer latent space representation? Please back up this statement.
- In the presentation of the ELBO, there is a typo after the second equality where p(z) is listed twice.
- This may just be personal preference on my behalf, but I think a short treatment on VAEs and optimal transport would be useful in the related work and background sections.
- Experimentally it would have been nice to have a direct comparison between this method and Mathieu et al. (2019), that paper being very similar to this one. At least a shared experiment would have been useful.







**Experience Assessment:**

I have read many papers in this area.

**Review Assessment: Checking Correctness Of Derivations And Theory:**

I assessed the sensibility of the derivations and theory.

**Review Assessment: Checking Correctness Of Experiments:**

I assessed the sensibility of the experiments.

**Review Assessment: Thoroughness In Paper Reading:**

I read the paper at least twice and used my best judgement in assessing the paper.

---

> ### Author Response · Authors · 2019-11-15
> **Authors' response**
>
> We thank the reviewer for the detailed feedback and would like to address the concerns below:
>
> Q: Section 3.2, please explain the significance of \lambda_x^c
>
> -> \lambda_x refers to the conformal factor between the hyperbolic and the Euclidean metric tensors $g_H$ and $g_E$
> respectively. There is actually a typo in the third equation: it should be $g_H = \lambda_x^2 g_E$
> The constant c describes the radius of the Poincaré ball and determines the degree of curvature of the
> hyperbolic space.
>
> Q: Please explain in the related work and/or introduce in more detail than you already have the differences and
> similarities between this work and your nearest neighbour “Hierarchical representations with poincaré variational
> auto-encoders” by Mathieu et al. (2019). I would like to know the advantages/disadvantages of using a Wasserstein formulation instead of a sampled ELBO.
>
> -> Typically the closed form version of the ELBO has significantly lower variance as opposed to the one-sample Monte
> Carlo ELBO gradient estimation. Based on this intuition, the choice was made to adopt the Wasserstein autoencoder
> approach (Section 4.4) since the Maximum Mean Discrepancy metric can be estimated using an unbiased minimum
> variance U-statistic estimator (Eq. (7)). The Maximum Mean Discrepancy metric can be estimated using a number
> of samples of the magnitude of the dimensionality of the latent space. While the MMD does not require parametric
> assumptions about the form of the distributions, allowing for more flexible encoders and decoders, it is known to
> perform well when matching high dimensional Gaussian distributions.
> Similar to the arguments presented in the original WAE paper, the use of kernel mean embedding via the Maximum Mean Discrepancy metric can be
> beneficial for the matching of the prior and the posterior due to the kernel properties. On the other hand,
> the selection of the kernel introduces further hyperparameters (the choice of the kernel and its bandwidth),
> which can be seen as a disadvantage. We will make sure to highlight these differences in the final revision of the paper.
>
> Q: Please explain in the related work and/or introduce in more detail than you already have the differences and
> In your section on “Dispersion representation” you state: “Since the maximum mean discrepancy can be estimated via
> samples, we do not require a closed form definition of the posterior density as is the case with training using the evidence lower bound.
> This allows the model to learn richer latent space representations.” Why exactly does not using a closed form posterior density
> lead to a richer latent space representation? Please back up this statement.
>
> -> The closed form of the Gaussian posterior in hyperbolic space allows for a scalar dispersion. Our model allows
> the use of vector dispersions (analogous to diagonal covariance in Euclidean space) as well as dense matrix dispersions
> since we perform the dispersion scaling in the tangent space as opposed to implicit reparametrization in Mathieu et al.
> We will make sure to present a more clear formulation of the statement in the final version of the paper.
>
> Q: Experimentally it would have been nice to have a direct comparison between this method and Mathieu et al. (2019),
> that paper being very similar to this one. At least a shared experiment would have been useful.
>
> -> Both Mathieu et al. and our work uses the MNIST dataset to demonstrate the latent space code distribution.
> We have also performed a number of more quantitative experiments on MNIST which expand on the reconstruction scores
> included in the appendix and will gladly include them in the revised version of the paper.
>
> Q: Novelty of the method
>
> -> As stated in Mathieu et al., their work has been developed concurrently with multiple authors, one of
> which is this work which has previously been published as a workshop paper.
>
> Q: Experimental results
>
> -> In our link prediction experiments, we show that the PWA model outperforms the Euclidean baseline
> on two of the datasets. In general, it is not trivial to determine which datasets could exhibit a latent space, hence an initial
> exploratory exposition is presented. Further work directed towards methods of identifying datasets with
> latent hierarchies could be a great avenue to explore.
>
> Q:
> - In section 3.3, please point readers forward to the Appendix for a list of gyrovector operations.
> - Please use equation numbers on all equations.
> - In the presentation of the ELBO, there is a typo after the second equality where p(z) is listed twice.
> - This may just be personal preference on my behalf, but I think a short treatment on VAEs and optimal transport would
> be useful in the related work and background sections.
>
> -> We will be happy to address these issues in the final revision.

---

### Official Review · AnonReviewer1 · 2019-10-22
**Official Blind Review #1**

**Rating:** 6

**Review:**

In this paper, the authors proposed a Poincare Wasserstein autoencoder for representing and generating data with latent hierarchical structures.
The proposed model extends the Wasserstein autoencoder to the hyperbolic space. It is a new and powerful member of the family of VAEs.
A hyperbolic Gaussian reparametrization method is designed and a Wasserstein loss with MMD regularizer is applied as an objective function.
The paper is well-organized and easy to read. The notations are clear.

Overall, I think this work is interesting. My main concern is about the experiments.
The results on MNIST is not sufficient to demonstrate the usefulness of the proposed method.
As the authors mentioned, a potential reason is that the MNIST is not a typical hierarchical dataset.
I suggest adding experiments on real-world hierarchical datasets, e.g., representing/generating sentences (which can be viewed as trees of words) or articles (which can be viewed as trees of sections/subsections).

**Experience Assessment:**

I have read many papers in this area.

**Review Assessment: Checking Correctness Of Derivations And Theory:**

I carefully checked the derivations and theory.

**Review Assessment: Checking Correctness Of Experiments:**

I carefully checked the experiments.

**Review Assessment: Thoroughness In Paper Reading:**

I read the paper at least twice and used my best judgement in assessing the paper.

---

> ### Author Response · Authors · 2019-11-15
> **Authors' response**
>
> We thank the reviewer for the feedback and will attempt to address the concerns in the following:
>
> Q: Overall, I think this work is interesting. My main concern is about the experiments.
> The results on MNIST is not sufficient to demonstrate the usefulness of the proposed method.
> As the authors mentioned, a potential reason is that the MNIST is not a typical hierarchical dataset.
> I suggest adding experiments on real-world hierarchical datasets, e.g., representing/generating sentences (which can be viewed as trees of words) or
> articles (which can be viewed as trees of sections/subsections).
>
> -> Of the three experiments we have provided in the paper, the intention behind the MNIST experiment was to shed light on some of the
> properties of the hyperbolic latent space in a qualitative way. We have also performed some quantitative experiments on MNIST which expand
> upon the reconstruction scores recorded in the appendix figures. We will make sure to include these in the final revision.
> The citation network datasets that are used in the link prediction exhibit partial hierarchical structure.
> The motivation for using the link prediction task as a benchmark was due to the fact that the model could capture
> some hierarchical properties of graphs in the latent space representations
> and leverage this capability in order to improve upon the Euclidean baseline scores.
> We agree with the suggestion that more experiments involving real-world hierarchical datasets could provide useful
> to illustrate the benefits of using the hierarchy inducing properties of the hyperbolic latent space.

---

### Official Review · AnonReviewer3 · 2019-10-23
**Official Blind Review #3**

**Rating:** 3

**Review:**

The work extend the variational autoencoder model to the hyperbolic space for exploring the hierarchical data structures. Experimental results on the synthetic and real data sets show the performance of the proposed model. In general, the paper is well organized.  The proposed algorithm is promising. My main concern is the motivation of the paper and the experiments. In particular,

(1) It is unclear what is the major contribution of this paper over the existing work, e.g., [10,22,23]. The authors may want to provide more discussion regarding this in the literature review.
(2) It is unclear where is the impact of the proposed PWA model in real applications. Also, it is unclear why the data like MNIST, CORA, CITESEER, PUBMED exhibit hierarchical data structure.
(3) The experiments are questionable. In table 2, why  CORA, CITESEER, PUBMED are adopted as the evaluation datasets? why only VGAE is considered as the baseline methods in link predict?

**Experience Assessment:**

I have published one or two papers in this area.

**Review Assessment: Checking Correctness Of Derivations And Theory:**

I carefully checked the derivations and theory.

**Review Assessment: Checking Correctness Of Experiments:**

I carefully checked the experiments.

**Review Assessment: Thoroughness In Paper Reading:**

I read the paper thoroughly.

---

> ### Author Response · Authors · 2019-11-15
> **Authors' response**
>
> We thank the reviewer for the feedback. In the following, we will attempt to address some of the concerns raised:
>
> (1) It is unclear what is the major contribution of this paper over the existing work, e.g., [10,22,23]. The authors may want to provide more discussion regarding this in the literature review.
>
> -> The biggest difference to existing work is the use of the Wasserstein autoencoder formulation and corresponding Maximum Mean Discrepancy metric to match
> the prior and posterior latent variable distributions. Although we do not explore the implications in our paper as the WAE formulation was
> a design choice in order to circumvent the large gradient variance associated with MC ELBO approximation, the benefits of adopting the MMD metric
> as regularizer include the better handling of outliers depending on kernel selection, which results in an improved
> overlap between the prior and posterior.
> Furthermore, our work uses a different reparametrization scheme which allows for more complex dispersion representations of the posterior.
> As stated in Mathieu et al., their work was developed concurrently to ours, which has previously been published as a workshop paper.
>
> (2 & 3) It is unclear where is the impact of the proposed PWA model in real applications. Also, it is unclear why the data like MNIST, CORA, CITESEER, PUBMED exhibit hierarchical data structure.
> The experiments are questionable. In table 2, why  CORA, CITESEER, PUBMED are adopted as the evaluation datasets? why only VGAE is considered as the baseline methods in link predict?
>
> -> We agree that more experimental results on real-world datasets such as text hierarchies would be beneficial to showcase the PWA model's capabilities.
> We explicitly state that we do not assume the MNIST to exhibit latent hierarchical structure. The motivation for the use of this dataset was mainly for expositional
> purposes to qualitatively demonstrate some of the properties of the hyperbolic latent space.
> The citation network datasets may partially exhibit hierarchical structure in their graphs. Similar to [5], where the
> partial cyclical structure of these datasets is explored, it is a reasonable benchmark to show whether the model can capture the
> partial hierarchical structure of the network graphs in the latent space codes which would show improvement upon the Euclidean VGAE
> baseline.
>
> [5] T. R. Davidson, L. Falorsi, N. De Cao, T. Kipf, and J. M. Tomczak. Hyperspherical variational
> auto-encoders.

---

### Decision · Program_Chairs · 2019-12-19

**Decision:**

Reject

**Comment:**

The paper received 3, 3, 6. All reviewers agree that the method is technically interesting. The main concern shared by the reviewers are the experiments which are somewhat underwhelming. The AC believes that this is a solid technical paper that needs a little bit more work. The authors are encouraged to strengthen their evaluation and resubmit to a future conference.